# Four-Dimensional Flow MRI for the Evaluation of Aortic Endovascular Graft: A Pilot Study

**DOI:** 10.3390/diagnostics13122113

**Published:** 2023-06-19

**Authors:** Paolo Righini, Francesco Secchi, Daniela Mazzaccaro, Daniel Giese, Marina Galligani, Dor Avishay, Davide Capra, Caterina Beatrice Monti, Giovanni Nano

**Affiliations:** 1Operative Unit of Vascular & Endovascular Unit, IRCCS Policlinico San Donato, Via Morandi 30, 20097 San Donato Milanese, Italy; daniela.mazzaccaro@grupposandonato.it (D.M.); dor.avishay@unimi.it (D.A.); giovanni.nano@unimi.it (G.N.); 2Department of Biomedical Sciences for Health, Università degli Studi di Milano, Via Mangiagalli 31, 20133 Milano, Italy; francesco.secchi@unimi.it; 3Unit of Radiology, IRCCS Policlinico San Donato, Via Morandi 30, 20097 San Donato Milanese, Italy; 4Magnetic Resonance, Siemens Healthcare GmbH, 91050 Erlangen, Germany; 5Postgraduation School in Radiodiagnostics, Università degli Studi di Milano, Via Festa del Perdono 7, 20122 Milano, Italy

**Keywords:** thoracic aortic endovascular repair (TEVAR), four-dimensional (4D) flow magnetic resonance, thoracic aortic dissection, aortic coarctation, helical flow, vortical flow, endoleak, computational fluid dynamic (CFD)

## Abstract

We aimed to explore the feasibility of 4D flow magnetic resonance imaging (MRI) for patients undergoing thoracic aorta endovascular repair (TEVAR). We retrospectively evaluated ten patients (two female), with a mean (±standard deviation) age of 61 ± 20 years, undergoing MRI for a follow-up after TEVAR. All 4D flow examinations were performed using a 1.5-T system (MAGNETOM Aera, Siemens Healthcare, Erlangen, Germany). In addition to the standard examination protocol, a 4D flow-sensitive 3D spatial-encoding, time-resolved, phase-contrast prototype sequence was acquired. Among our cases, flow evaluation was feasible in all patients, although we observed some artifacts in 3 out of 10 patients. Three individuals displayed a reduced signal within the vessel lumen where the endograft was placed, while others presented with turbulent or increased flow. An aortic endograft did not necessarily hinder the visualization of blood flow through 4D flow sequences, although the graft could generate flow artifacts in some cases. A 4D Flow MRI may represent the ideal tool to follow up on both healthy subjects deemed to be at an increased risk based on their anatomical characteristics or patients submitted to TEVAR for whom a surveillance protocol with computed tomography angiography would be cumbersome and unjustified.

## 1. Introduction

From its introduction, four-dimensional (4D) flow magnetic resonance imaging (MRI) has been considered the in vivo reference standard for non-invasive hemodynamic assessment [1]. The flow evaluation used when 4D flow is unavailable is based on computational analysis (computational fluid dynamics, CFD). Based on the computed tomography angiography anatomy, this technique allows to create a flow inside a vessel. A 4D flow is performed using spoiled gradient-echo sequences with a short repetition time (TR), electrocardiographic, and respiratory gating. Due to interindividual variability in heart rate and breathing patterns, total acquisition time can range from 5 to 15 min [2]. Using retrospective, rather than prospective, gating enables near-full coverage of the cardiac cycle. Data acquisition is synchronized with the cardiac cycle, and data collection is distributed over multiple cardiac cycles. After the completion of the 4D flow acquisition, four time-resolved (CINE) 3D datasets are generated, and it is possible to obtain a 3D reconstruction of vascular structures [3].

With this technique, it is possible to assess a volumetric blood flow over the entire vessel of interest with volumetric quantification and retrospective analysis of blood flow through any plane [2]. Blood velocities exceeding the velocity encoding (VENC) result in velocity aliasing, precluding flow measurements. VENC is usually set at 10% above the expected maximum velocity. However, high VENC increases noise and decreases the velocity-to-noise ratio, especially in regions of low velocity [3]. A 4D flow can be performed without a contrast agent. The use of contrast agents significantly improves the signal-to-noise ratio in magnitude data and noise reduction in velocity data compared to measurements without contrast agents [3]. Using 3D or 4D PC MR angiography derived from 4D flow MRI data helps with anatomic orientation and identification of cross-sectional analysis planes for flow quantification. A 4D flow was used in the assessment of several clinical conditions, from aortic valve diseases [4] to complex congenital heart pathologies [5]. In all such cases, a 4D flow allows both a qualitative assessment of blood flow dynamics, deriving from the encoded velocities parameters, such as jet angles [6], and wall shear stress [7]. Wall shear stress refers to the stress applied tangentially to the vessel wall, that is, the tangential viscous shear forces per unit area exerted by the shear in the fluid layer immediately adjacent to the wall. Another important measurement obtained with a 4D flow is a pulse-wave velocity. The pulse-wave velocity is the velocity of the pulse-wave propagation along a vessel, usually an artery, and it is a marker of arterial stiffness and predictive of cardiovascular disease.

In vivo morphometrical analyses that quantify remodeling of the aging human thoracic aorta, both healthy and diseased, represent an important field of study with several potential clinical applications. Thereafter, all the parameters and information derived from the 4D flow sequences analysis could be useful for investigating the aortic flow modifications, both in the normal aorta and in the presence of aortic diseases, but also after treatment. In this regard, 4D MRI has been reported as a useful diagnostic tool to characterize the complex hemodynamic and flow patterns seen in the normal aorta of patients of different ages and therefore having an increased grade of aortic tortuosity [8].

The potential role of a 4D flow in the follow-up of endovascular aortic repair, either of abdominal or thoracic aortic segments, has been explored. Indeed, thoracic endovascular aortic repair (TEVAR) requires lifelong postoperative surveillance, as it could be affected by long-term complications, such as endoleaks [9], which are defined as a persistent blood flow outside the stent and within the vessel walls, stent collapse, stent infection, and stent migration. A flow analysis utilizing 4D Flow MRI has been applied as a diagnostic tool in type B aortic dissection in which analyzing the flow through the primary entry tear from the native true lumen into the false lumen may help to predict adverse outcomes and aid in risk-stratifying patients for pre-emptive surgical/endovascular procedures [10,11]. The mechanical coupling between thoracic aortic stent grafts and the aorta represents a main issue in the follow-up of patients treated with TEVAR, and requires more exploration and optimization to improve the clinical outcomes, especially in the case of the patients submitted to an endovascular procedure for acute type B aortic dissection. Studies based on 4D Flow MRI could better clarify qualitative assessment of blood flow dynamics with an in vivo evaluation of flow patterns, examine potential links between proximal geometrical characteristics of the thoracic aorta and distal TEVAR complications, such as distal stent graft-induced new entry (dSINE), or proximal TEVAR complications related to distal geometrical characteristics of the thoracic aorta. Furthermore, computational fluid dynamics studies showed how hemodynamic factors could exert a relevant drag force on the stent, possibly leading to stent migration or failure [12]. Initially, Hope et al. [13] first proposed the use of the 4D flow for the identification of endoleaks after endovascular aneurysm repair (EVAR) in a case report, and then, Rengier et al. [14] and Bunk et al. [15] demonstrated the feasibility of in-stent flow visualization in phantom studies, establishing that the stent presence does not undermine flow measurements. More recently, Sakata et al. [16] characterized each type of endoleak after EVAR by its flow appearance, describing a superior sensitivity to computed tomography angiography in endoleak detection. Subsequently, the same team investigated the predictive power of a 4D flow analysis of type 2 endoleaks, suggesting that it could be useful for the prediction of sac expansion after EVAR [17]. Furthermore, Ravesh et al. [18] reported how a 4D flow allowed the assessment of relevant hemodynamic parameters, namely, blood flow velocity and wall shear stress in a patient with a folded endograft following TEVAR.

The purpose of this paper is to explore the feasibility of 4D flow MRI for patients undergoing aortic endovascular grafting.

## 2. Materials and Methods

### 2.1. Study Population

We retrospectively evaluated 10 patients (2 female), with a mean (±standard deviation) age of 61 ± 20 years, undergoing MRI for follow-up after TEVAR in our center. All patients underwent TEVAR except one, who had an endovascular stent therapy for isthmic coarctation with a covered CP Sent (CCPS; NuMED, Inc., Hopkinton, NY, USA).

### 2.2. Image Acquisition

All 4D flow examinations were performed using a 1.5-T system (MAGNETOM Aera, Siemens Healthcare, Erlangen, Germany) using a 48-channel surface phased array coil, placed over the thorax of the patient in a supine position. In addition to the standard examination protocol, a 4D flow sensitive 3D spatial-encoding, time-resolved, phase-contrast prototype sequence (WIP 785B) with retrospective electrocardiogram gating and respiratory gating was acquired. This package supports the acquisition of volumetric phase contrast data with velocity vector encoding. Acquisition parameters were: TE 2.3–3.1 ms, echo spacing 5.1 ms, flip angle 8°, segment number 2, temporal resolution 40.6–43.4 ms, bandwidth 490 Hz/pixel, FOV 340–232 mm^2^, 3D acquired resolution 3.5 × 2.4 × 3.9 mm, and VENC 100 cm/s. Images were acquired in a sagittal plane, yielding 40 to 52 slices depending on the patient’s size.

### 2.3. Image Analysis

For the post-processing, we used a 4D flow prototype Demonstrator Version 2.4, a prototype software package that allows the qualitative and quantitative analysis of blood flow using time-resolved phase-contrast MRI datasets. For each patient, we estimated the flow rate (mL/beat) at three different points: ascending aorta (A1), proximal part of endovascular prosthesis (A2), and in a plane immediately below the endovascular prosthesis (A3). Finally, we also evaluated the presence of artifacts.

### 2.4. Statistical Analysis

Data are reported as the median and interquartile range (IQR) due to the small sample size. Potential differences between flows were appraised. Such differences were analyzed using the Mann–Whitney U test for continuous variables and with Fisher χ^2^ for categorical variables. Statistical analyses were performed with R 3.5.2 (R Foundation for Statistical Computing, Vienna, Austria), and *p*-values ≤ 0.05 were considered indicative of statistical significance.

## 3. Results

The median flow rate was 75 mL/beat in A1 (interquartile range 47–93), 54 mL/beat in A2 (interquartile range 40–59), and 50 mL/beat in A3 (interquartile range 35–55). Values decreased from the ascending aorta to the proximal endoprosthesis due to the branching of supra-aortic vessels with a significant difference (*p* = 0.043). Values decreased from A2 to A3 without a significant difference (*p* = 0.326) due to TEVAR (Table 1).

Among our cases, flow evaluation was feasible in all patients, although in 3 out of 10 patients, we observed some artifacts due to the metallic structure of the endograft. Indeed, three individuals displayed a reduced signal within the vessel lumen where the endograft was placed, while others presented with turbulent or increased flow.

Concerning patients with blood flow turbulence, we observed a vortical flow in the distal portion of the ascending aorta of one 60-year-old male patient that underwent TEVAR for sub-acute Type B aortic dissection (Figure 1). This patient showed an accelerated flow right above the aortic valve in peak systole, contributing to the generation of a vortex and velocity of flow towards the late systolic phase. The accelerated and vortical flow was situated at the proximal sealing zone of the endograft, and thus the vortex could be attributed to the shape of the inferior wall of the traverse aortic arch, which contributed to forming an angle together with the different grades of change of distensibility of the aortic wall compared to the endograft during the cardiac circle.

Three patients presented an aortic flow acceleration. A 70-year-old male patient treated with off-pump arch debranching followed by TEVAR in zone 0 for aortic aneurysm post chronic type B dissection showed an helical flow both in distal ascending aorta and isthmic aorta, likewise acceleration of the aortic flow in the distal section of the endograft (Figure 2 and Appendix A).

Two other cases are presented: a 53-year-old male patient with repaired dissection displayed an accelerated flow, and a 51-year-old male patient with a repaired aneurism of the aortic arch presented flow accelerations in correspondence with endograft proximal and distal endo-anastomosis (Figure 3).

Concerning reduced flows, in one case (Figure 4), a 10-year-old female treated with a CP-covered stent (CCPS; NuMED, Inc., Hopkinton, NY, USA) for aortic coarctation, a flow reduction may be observed where the aortic lumen tightens. This may be due to the fact that the narrowing causes an acceleration of the flow over the velocity encoding, which is set a priori, or to alterations in the magnetic field induced by the stent. However, an acceleration of the blood flow is visible in the aortic arch due to the angulation of the arch. In the second and third cases, an 82-year-old female and an 81-year-old male with TEVAR presented a reduced flow in the aortic lumen, likely attributable to a difficult ECG trigger or stent-induced artifacts.

Nevertheless, in all the other patients, the flow signal magnitude was unaltered throughout the graft lumen, granting a complete representation of the flow. For instance, in a 53-years-old patient that underwent TEVAR after a traumatic aortic rupture, the flow was laminar in all the visualized aortic segments, acceleration was visible only at the aortic valve level, as expected in normal conditions, and no flow alterations were observed inside the endograft and after the endograft (Figure 5).

## 4. Discussion

Nowadays, computed tomography angiography imaging is considered the reference standard for the post-operative follow-up of TEVAR. Nevertheless, 4D flow MRI offers the unique advantage of a comprehensive analysis of blood flow in the repaired aorta through in vivo flow evaluation, without the need for contrast agents. Non-contrast MRI can be performed to monitor patients treated with TEVAR, offering high sensitivity [19]. The ability to visualize different flow patterns and derive dynamic parameters, such as wall shear stress, might provide new insight into the risk of post-operative complications. Indeed, computational fluid dynamics revealed that aortic segments presenting low wall shear stress are linked to thrombus formation, whereas increased wall shear stress is related to the propagation of aortic dissection [20]. The limit of this approach is that computational fluid dynamics is only a flow simulation, whereas the 4D flow MRI may allow an in vivo assessment of such features without radiation exposure or administration of contrast agents.

In our patient submitted to TEVAR for sub-acute Type B aortic dissection, 4D MRI was able to evaluate in vivo changes of blood flow and wall share stress in the distal part of the ascending aorta during the late systolic phase, demonstrating a vortical and helical flow. This finding is particularly interesting in the context of the proximal landing zone of an endograft in zone 1, but mainly in zone 2 or 3 of Ishimaru in the type III aortic arch [21,22,23]. In this configuration, the lesser curvature of the traverse arch and the undersurface of a thoracic endograft form a wedge-shaped gap known as the “bird beak” configuration (Figure 6). This configuration, characterized by the angle (α angle) between the undersurface of the endograft and the aortic wall and the length of the protruding longitudinal segment (PLS) of the endograft, indicates incomplete apposition of the endograft to the lesser curvature of the arch [24]. The α angle indicates an incomplete apposition of the proximal edge of the endograft to the lesser curvature of the arch. [25] As a result, an insufficient proximal seal or endograft migration or both may occur, leading to a type Ia endoleak with the inherent continued risk of late TEVAR complications such as migration, infolding, or even aortic rupture [26]. Investigation of an in vivo helical aortic flow using a 4D flow in this setting, obtaining information about the presence, overall magnitude, and direction of rotation of the helical flow in the bird beak appear helpful in detecting helical patterns, particularly the risk for long-term TEVAR complication.

Moreover, the intra-aortic blood flow pattern regionally assessed with computational fluid dynamics through the Modified Arch Landing Areas Nomenclature (MALAN) showed a specific, consistent, and abnormal secondary helical flow pattern, which may account for its high prevalence in patients with type B AD [27]. Specifically, in zone 3 (i.e., the isthmus), which identifies the most common site for proximal entry tear in type B AD, type III arch presents a high rotational helical flow, heavily insisting on the aortic wall of that vessel segment (i.e., MALAN area 3/III) [27]. Four-dimensional flow resonance could become a potential new method to evaluate and monitor in vivo exacerbation of a helical flow in patients affected by type B AD, as it associates with both the onset and the evolution of the disease. Moreover, the possibility of evaluating the presence and degree of a helical flow could represent a new target for anti-hypertensive therapy to avoid the onset of type B AD or the aneurysmal evolution of the false lumen. In particular, this could include hypertensive patients with aortic arch configurations at higher risk for acute aortic syndromes (i.e., MALAN area 3/III), or patients already affected by chronic type B AD. Likewise, as reported in computational fluid dynamic modeling studies, which highlight the increase in the magnitude and direction of the displacement forces for each type of aortic arch geometry using MALAN classification [22,23], 4D flow MRI was able to show in vivo vortical flow and helical flow in zone 2 and 3 of type III aortic arch (zones 2/III and 3/III MALAN classification), despite the presence of the endograft in patients submitted to off-pump arch debranching followed by TEVAR in zone 0 for aortic aneurysm post chronic type B aortic dissection. Recently, Dyverfeldt et al. demonstrated with a 4D flow MRI study that tortuosity increases with age and blood flow in tortuous aortas are more helical [28]. From this perspective, the possibility of highlighting in vivo vortical and helical flow patterns by 4D flow MRI in Ishimaru zones 1, 2, and 3 in patients undergoing TEVAR provides new perspectives for both planning the procedure in a pre-op phase and for the follow-up of these patients who require long periods of clinical monitoring. Therefore, 4D flow MRI may represent the ideal tool to follow up with both healthy subjects deemed to be at an increased risk, based on their anatomical characteristics, or patients submitted to TEVAR for whom a surveillance protocol with computed tomography angiography would be cumbersome and unjustified. On the other hand, with computed tomography angiography, it is possible to evaluate other new parameters, such as sarcopenia, which could be useful to identify patients with a lower long-term survival rate, irrespective of the patient’s age or gender [29,30].

Currently, the management of type II endoleaks is decided upon the growth of an aneurismatic sac [31]. Computed tomography angiography is commonly used for post-EVAR follow-up in type I endoleak detection and type II endoleak surveillance because of its rapid acquisition time and high diagnostic value [30]. However, the superiority of MRI for detecting endoleaks has been previously reported. In this regard, the 4D character of acquisition (three spatial and one temporal dimension) does provide additional hemodynamic information about the endoleak [32]. Regarding endoleak analysis, Hope et al. previously studied type I endoleaks by using a 4D flow [13]. However, type II endoleaks have been less investigated, possibly because of the difficulty of assessing the aortic side branches with a 4D flow. Nowadays, a 4D flow analysis of endoleak flow patterns, such as flow direction, volume, and velocity might allow more timely detection of endoleak complications leading to more accurate and tailored treatment approaches [16]. On the other hand, a 4D flow analysis may be associated with higher false-positives in endoleak detection [16].

Several recent studies have examined the feasibility of studying a 4D flow in transcatheter aortic valve implantation. Aigner et al. concluded that utilizing 4D MRI techniques, which allow for qualitative flow assessment in a patient-specific, MR-compatible, and flexible model, was achievable through 3D printing techniques [33]. An in vivo study demonstrated flow modifications in the ascending aorta following transcatheter aortic valve implantation, particularly highlighting an asymmetric distribution of systolic wall shear stress in the ascending aorta [34,35]. The potential introduction of MR 7T offers a future perspective, providing a new tool for exploring aortic 4D flow. This advancement would yield a higher signal-to-noise ratio, allowing to push the boundaries in terms of acceleration and resolution, and significantly improving the overall signal-to-noise ratio [36].

## 5. Limitations

Our study has some limitations that prevents generalization. First, the study was a single-center study with a limited number of patients. Second, the heterogeneity of the population does not permit to give a definitive conclusion. However, it is impossible to acquire a 4D flow in all the exams performed due to the long acquisition time. Prospective studies are necessary to assess the prognostic value of a 4D flow for the follow-up after aortic endovascular treatment.

## 6. Conclusions

The presence of an aortic endograft does not necessarily impede the visualization of blood flow through 4D flow sequences. However, it is important to note that flow artifacts may be generated by the graft in some cases. Integrating a hemodynamic evaluation into the routine clinical practice of aortic endovascular treatments could potentially offer valuable insights into the flow changes that occur after TEVAR, shedding light on their impact on the procedure’s success. In this context, 4D flow MRI may serve as an ideal tool for monitoring both healthy subjects who are deemed to have an increased risk based on their anatomical characteristics and patients who have undergone TEVAR, as utilizing a surveillance protocol involving computed tomography angiography would be cumbersome and unjustified.

## Figures and Tables

**Figure 1 diagnostics-13-02113-f001:**
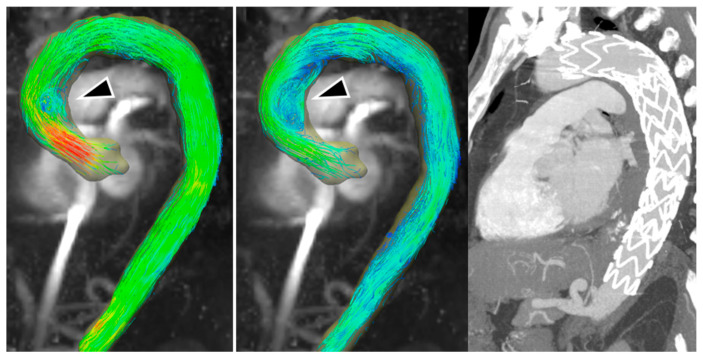
TEVAR used to treat sub-acute type B aortic dissection in a 60-year-old male. Maximum intensity projection image of flow speed, 3D streamlines visualization. (**Left**) Peak systolic phase. (**Middle**) Mid-systolic phase showing a vortex forming in the distal portion of the ascending aorta (black arrowhead). (**Right**) Maximum intensity projection image from computed tomography scan.

**Figure 2 diagnostics-13-02113-f002:**
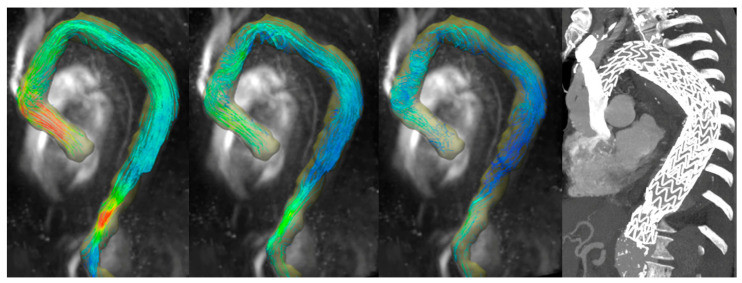
A 70-year-old male patient treated with off-pump arch debranching followed by TEVAR in zone 0 for aortic aneurysm post chronic type B dissection. From the left, a maximum intensity projection image of flow speed with 3D streamlined visualization is shown. An increased velocity in the distal part in the overlapping zone of the two endografts and shape of the aorta causing a slight narrowing of the lumen is shown. On the right, a computed tomography scan demonstrated the position of the endograft.

**Figure 3 diagnostics-13-02113-f003:**
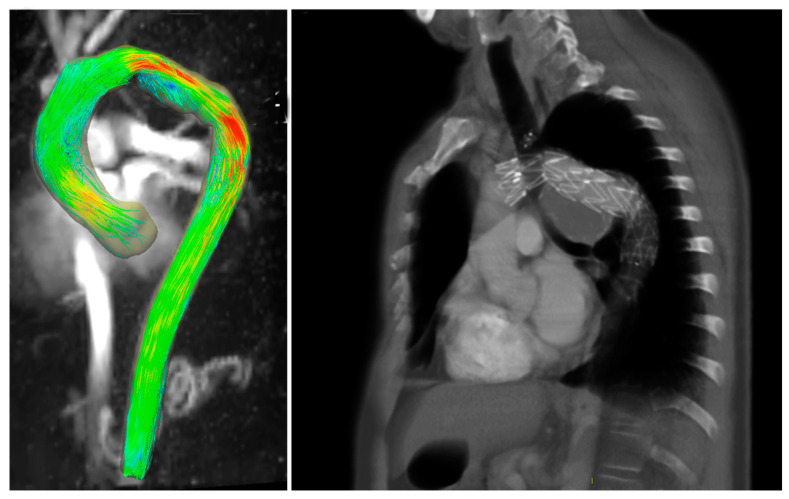
A 51-year-old male patient with TEVAR. (**Left**) Maximum intensity projection image of flow speed. (**Right**) Computed tomography scan showing the position of the endograft.

**Figure 4 diagnostics-13-02113-f004:**
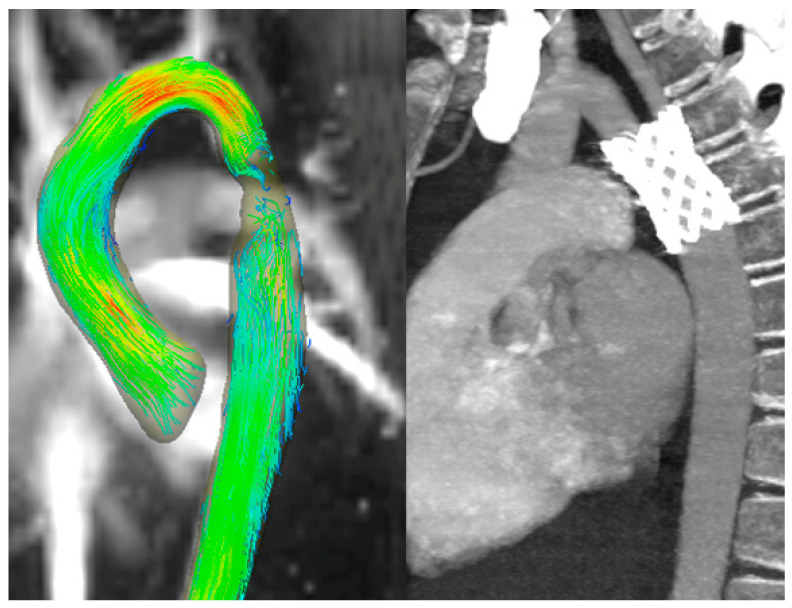
A 10-year-old female patient treated for aortic coarctation at isthmus level. (**Left**) Maximum intensity projection image of flow speed, 3D streamlines visualization. (**Right**) Maximum intensity projection image from computed tomography scan is shown.

**Figure 5 diagnostics-13-02113-f005:**
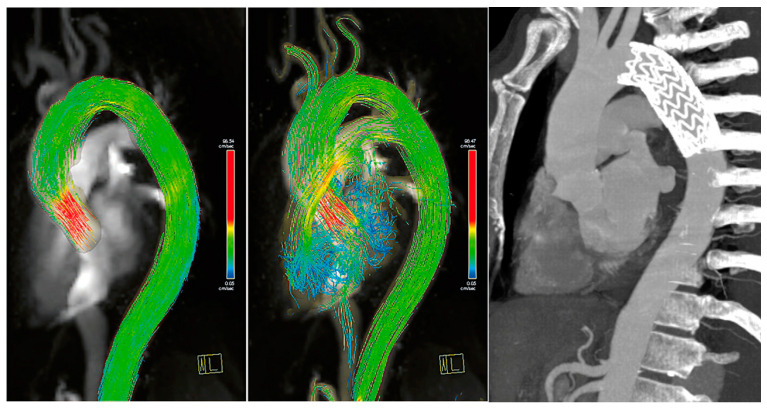
A patient treated with endograft for traumatic aortic rupture. (**Left** and **middle**) Maximum intensity projection image of flow speed with 3D streamlined visualization. Note the higher velocity in the ascending aorta and flow velocity is also valuable in an endovascular graft. (**Right**) Computed tomography scan showing the position of the endograft.

**Figure 6 diagnostics-13-02113-f006:**
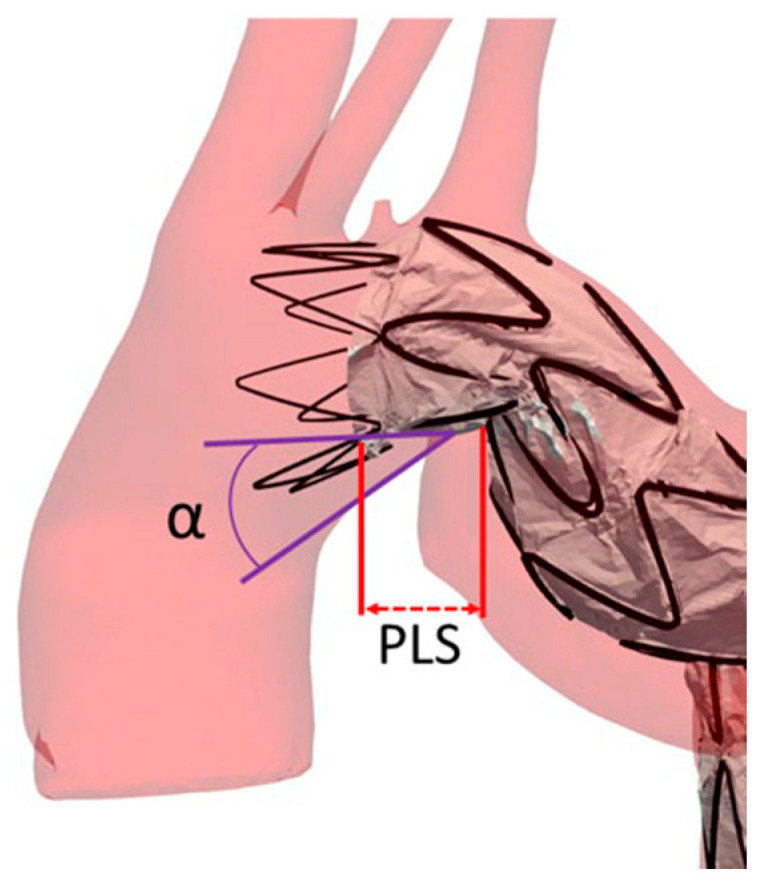
Bird-beak configuration specific features: the angle between the undersurface of the endograft and the aortic wall (α angle), and the length of the protruding longitudinal segment (PLS) of the unopposed stent-graft.

**Table 1 diagnostics-13-02113-t001:** The table shows values decreasing from A1 (ascending aorta) to A2 (proximal endoprosthesis) with a significant difference (*p* = 0.043) and from A2 to A3 (distal endoprosthesis) without a significant difference.

	Flow Rate mL/Beat	*p* Value
A1	75 (47–93)	0.043 (A1 vs. A2)
A2	54 (40–59)	0.326 (A2 vs. A3)
A3	50 (35–55)	0.044 (A1 vs. A3)

## Data Availability

Data is available upon reasonable request to the corresponding author.

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
