# Peer review of "Four-Dimensional Flow MRI for the Evaluation of Aortic Endovascular Graft: A Pilot Study"

_diagnostics, 2023, doi:10.3390/diagnostics13122113_

Round 1
Reviewer 1 Report
1. Introduction is too long. Much of this could be covered in the discussion. The authors should revise the introduction to get more quickly at the point of the paper.
2. Results, page 4, line 182: What was the nature of the artifact in three patients? Since all patients had metal stents, I would think the artifact would be more or less the same in everyone.
3. Results, page 5, line 186: What is the clinical significance of vortical flow?
4. Results, page 5, line 201: Similarly, what is the clinical significance of flow acceleration?
5. Discussion: With respect to type B dissections, are there MRI/flow dynamic findings that would suggest a more complicated course of the dissection? Are there any predictive findings that would help guide conservative for endovascular therapy of acute/subacute uncomplicated dissections?
6. Discussion: Along similar lines, could the authors discuss practical clinical applications of this technology. How does one incorporate flow dynamics into clinical decision making?
7. Discussion: What is the scalability of this technology? A CT scan is pretty quick and simple to obtain. It would seem that the longer acquisition and processing times of 4D MRI would make it much more difficult to incorporate into daily practice. Who are the patients one should consider using this technology for?
Manuscript will require English language editing
Author Response
Dear Editor, Dear Reviewers,
We thank you for the effort spent in the revision of our reviewed manuscript. We have revised the paper according to your valuable suggestions modifying part of text and answering to your questions. We believe that now manuscript has been improved by your suggests.
Once again, thank you for your precious time.
On behalf of all Authors.
Reviewer #1:
- Introduction is too long. Much of this could be covered in the discussion. The authors should revise the introduction to get more quickly at the point of the paper. Reply: Thank for this observation. We moved some parts of the introduction in the discussion section.
- Results, page 4, line 182: What was the nature of the artifact in three patients? Since all patients had metal stents, I would think the artifact would be more or less the same in everyone. Reply: we define better the nature of the artifact.
- Results, page 5, line 186: What is the clinical significance of vortical flow? Reply: the presence of vortical flow is correlated to the increase of aortic diameters in particular in patient with bicuspid aortic valve (see ref #7).
- Results, page 5, line 201: Similarly, what is the clinical significance of flow acceleration? Reply: the flow acceleration could be related to a stenosis or a curvature of aorta or endograft.
- Discussion: With respect to type B dissections, are there MRI/flow dynamic findings that would suggest a more complicated course of the dissection? Are there any predictive findings that would help guide conservative for endovascular therapy of acute/subacute uncomplicated dissections? Reply: thanks for this observation. There are no data regarding this aspect in type B dissection. On the bases of a study that showed consistent esacerbation of helical flow in zone 3 of type II arch (MALAN 3/III) we believe that with 4D analysis in the future will be possible to have some predictive findings. We add a comment in the text about this topic. (See P 8 L492-502)
- Discussion: Along similar lines, could the authors discuss practical clinical applications of this technology. How does one incorporate flow dynamics into clinical decision making? We thank the Reviewer for the valuable comment. We add a comment in the text about this topic (See P 8 492-502)
- Discussion: What is the scalability of this technology? A CT scan is pretty quick and simple to obtain. It would seem that the longer acquisition and processing times of 4D MRI would make it much more difficult to incorporate into daily practice. Who are the patients one should consider using this technology for? Reply: of course in this moment we use 4D flow only for scientific purpose due to the long time of acquisition compared to CT but with the recent technology development (i.e. new sequences or 3T MRI) we believe that in the future 4d flow will be use in some clinical settings to obtain more diagnostic and prognostic information.
Reviewer 2 Report
The manuscript entitled `Four-dimensional Flow MRI for the Evaluation of Aortic Endovascular Graft: a Pilot Study` written by Paolo Righini et al. presents the application of 4D MRI to the evaluation of patients after thoracic aorta endovascular repair (TEVAR). The authors showed benefits gained from using this technique, especially in situations with turbulent, increased or decreased flow and in the presence of artifacts. The presented results could contribute to broadening the use of 4D MRI imaging to provide better care of patients after TEVAR. The text is well structured and clear, scientifically sound, and the references were selected properly. I have only a few minor comments for the authors:
1 1. In the Introduction, the description of 2D methods was provided, following with the characterization of 4D techniques. It raises the need for mentioning of 3D imaging methods in the context of 2D and 4D techniques.
2. In the legend to Figure 2, panels A, B, and C are mentioned, but such a labelling is not present in this figure. Please check this.
I believe that my suggestions will be helpful to the authors to increase the quality of the reviewed work.
The text contains minor editorial errors, please check lines 18, 107, and 129. Although such errors will surely be identified by publisher staff at a later stage of the manuscript processing, I would like to address them out now.
Author Response
Dear Editor, Dear Reviewers,
We thank you for the effort spent in the revision of our reviewed manuscript. We have revised the paper according to your valuable suggestions modifying part of text and answering to your questions. We believe that now manuscript has been improved by your suggests.
Once again, thank you for your precious time.
On behalf of all Authors.
Reviewer #2:
- In the Introduction, the description of 2D methods was provided, following with the characterization of 4D techniques. It raises the need for mentioning of 3D imaging methods in the context of 2D and 4D techniques. Reply: thanks of this observation. We add a sentence regarding the possibility to obtain a 3D reconstruction of aorta using 4D floe seqences.
- In the legend to Figure 2, panels A, B, and C are mentioned, but such a labelling is not present in this figure. Please check this. Reply: we modify the figure accordingly.